REGISTERED REPORT PROTOCOL

# The impact and return-on-investment of evidence-based practice in conservation and environmental management: A machine learning-assisted scoping review protocol

Alec P. Christie[1,2]*, Philip A. Martin[3,4], Nigel G. Taylor[2]

1 Centre for Environmental Policy, Imperial College London, London, United Kingdom, 2 Department of Zoology, University of Cambridge, Cambridge, United Kingdom, 3 Basque Centre for Climate Change (BC3), Leioa, Bizkaia, Spain, 4 IKERBASQUE—Basque Foundation for Science, Bilbao, Bizkaia, Spain

* a.christie@imperial.ac.uk

## Abstract

Evidence-based Practice (EBP) is a vital principle, with its origins in the 1970s, that has transformed the disciplines of medicine and healthcare. The use of best available evidence to inform decisions and best practice has since spread across other disciplines, including in the environmental sciences through evidence-based conservation and environmental management. However, ironically there only appears to be a single scoping review on the impacts and return-on-investment of EBP in healthcare and it is unclear whether any such evidence exists in the broad field of conservation and environmental management. In this scoping review, we aim to explore the extent to which evaluations of the impacts and return-on-investment of EBP and evidence use have been conducted in conservation and environmental management on both human and environmental outcomes. We will search at least ten different electronic bibliographic platforms, databases, and search engines for published and grey literature, from 1992 to 2025 – there will be no geographical or language restrictions on the documents included. A machine learning-assisted review process will be followed using open source tools (ASReview and SysRev) and following the comprehensive SYstematic review Methodology Blending Active Learning and Snowballing (SYMBALS). The findings from the scoping review will be useful to inform organisations and practitioners considering implementing EBP on its benefits and costs and will also highlight potential research gaps on the impact of EBP and evidence use.

## 1. Introduction

The principle of Evidence-Based Practice (EBP), by that name, emerged in medicine and healthcare around 1992 [1] but has its origins earlier in the 1970s with Archie Cochrane's article on 'Effectiveness and efficiency' in health services [2]. Put simply,

**Data availability statement:** All relevant data from this study will be made available upon study completion.

**Funding:** Alec Christie was funded by Imperial College London through an Imperial College Research Fellowship.

**Competing interests:** The authors have declared that no competing interests exist.

EBP has been defined as "a systematic process to underpin actions and decisions with the best available evidence" [3]. A consensus definition of EBP was later developed in the Sicily statement on evidence-based practice as: "EBP requires that decisions about health care are based on the best available, current, valid, and relevant evidence. These decisions should be made by those receiving care, informed by the tacit and explicit knowledge of those providing care, within the context of available resources." [4] Therefore, the general goals of EBP and evidence-based decision-making (EBDM) are to enhance the quality of interventions, improve intervention outcomes, and reduce unwarranted variations in practice that lead to inefficiencies, and reduce costs [5]. After several decades, EBP has become a wider multi-disciplinary pursuit, aiming to increase the effectiveness and efficiency of practice across fields, with the term EBP evolving and shaping other disciplines, such as through the formation of concepts such as evidence-based policing, evidence-based education, and evidence-based conservation.

Evidence-based practice appears to have first been introduced to the environmental literature in the early 2000s [6–10] within the field of Evidence-based Conservation. The key goals of Evidence-Based Conservation (EBC) were to improve the effectiveness of conservation (i.e., minimising harmful and ineffective outcomes and maximising beneficial ones) to stop and reverse biodiversity loss with limited resources [11–13]. The EBC paradigm draws heavily from the concept of EBP, and specifically Evidence-Based Medicine (EBM) [1,3], and seeks to reduce the influence of subjective opinions, biases, and unfounded beliefs on conservation decisions. This was motivated by research that found that decisions in conservation practice were largely based upon personal experience and opinion without considering scientific evidence, thus hampering effective conservation practice [14,15]. Since then, proponents of EBC have often criticised those implementing conservation practice and policy for 'evidence complacency' [16] and criticised poor use of scientific evidence in management plans and decision-making [17]. The push for more EBC has led to several large synthesis projects, such as Conservation Evidence and the Collaboration for Environmental Evidence [18–20] that have worked within this research-implementation space [21] between scientists, practitioners, and policymakers to improve evidence-based environmental decision-making.

Whilst substantial progress has been made on collating scientific evidence in practitioner-friendly forms (e.g., the Conservation Evidence database [19]), EBP is not yet widely adopted across the conservation and environmental management sectors and may even be resisted by some decision-makers [16,22,23]. For example, one criticism of EBP in the past has been that there is too much focus on scientific evidence and not other forms of evidence that exist (despite the inclusion of 'tacit and explicit knowledge' in the Sicily Statement definition above). It is argued that these other forms of knowledge, such as local, traditional, practitioner, and indigenous knowledge, might also be useful for informing practice and policy for conservation and environmental management [24–26]. As a result, more recently, a broader definition of evidence has been proposed and used in conservation and environmental management compared to EBM [27,28]. This will be used for the purposes of this review,

as follows: "any relevant data, information, knowledge, and wisdom used to assess an assumption, claim, or hypothesis related to a question of interest". This includes a plurality of sources of evidence such as an evidence synthesis, a primary peer-reviewed research paper, grey literature report, practitioner, local, indigenous and/or expert knowledge, observations, and experience [27,28]. It is also important to acknowledge that the term 'evidence-informed' conservation has also been proposed to account for other forms of knowledge and the need to consider values when making decisions – however, this is debated and both terms are still used [24,29–31].

Nevertheless, there are still many other significant barriers [32] to implementing EBP to protect, conserve, and restore the environment. One such problem is that the types of experimental and quasi-experimental study designs used in conservation differ greatly to those in medicine and other fields, tending to be less rigorous and more prone to bias as it is challenging to practically or ethically conduct randomised experiments in conservation [33–38].

A further key challenge is that implementing EBP requires practitioners and organisations to redirect time and resources away from direct action. For example, spending time consulting evidence potentially takes time and/or resources away from implementing direct interventions to save species from anthropogenic threats. This is particularly problematic in conservation, which typically has scarcer resources compared to medicine, whilst at the same time facing rapid environmental changes and species declines. Of course, it is known that delaying action can sometimes be a cost-effective strategy [39]. However, to convince practitioners and organisations of the value and need for EBP in the environmental sector, we must be able to demonstrate that EBP is not harmful to conservation efforts and represents a worthwhile investment – for example, delaying action to review the evidence base to eradicate an invasive species could facilitate its spread and increase overall costs [40]. Indeed, proponents of EBP often neglect to recognise that there are two key underlying assumptions of EBP, which are that interventions based on consulting evidence: 1. result in better outcomes; and 2. are associated with reduced costs. Together, this means that there is a positive return-on-investment (ROI) or value-on-investment (VOI) from implementing EBP. However, it is rather ironic that, to the best of our current knowledge of the literature, that the first review on the outcomes of EBP and its return-on-investment was only published in 2023 in healthcare [41]. The scoping review found that primary studies reported mixed effects of EBP on relevant patient outcomes, but overall there were benefits to patient outcomes and that based on the limited available ROI data (from 19% of studies found), most studies (94%) reported that EBP has a positive ROI [41]. Whilst these results are encouraging for healthcare, it is unclear whether the same applies to EBP in other disciplines, specifically evidence-based conservation and environmental management.

It is appropriate to tackle this key knowledge gap with a scoping review given the general lack of understanding as to whether any substantial amount of literature may exist on the impacts and ROI or VOI of EBP outside of healthcare. Specifically, the scoping review planned here aims to explore the extent to which evaluations of the impacts and ROI or VOI of EBP have been conducted in the fields of conservation and environmental management on both human and environmental outcomes.

## 2. Materials and methods

We report this scoping review protocol according to the PRISMA Extension for Scoping Reviews [PRISMA-ScR]: Checklist and Explanation [42] and will conduct the final review in line with this protocol – we will detail any changes in the final review. Furthermore, in the development and planning of this protocol we were guided by the Joanna Briggs Institute (JBI) approach and the Population (or participants)/Concept/Context (PCC) framework, which is the most updated and rigorous approach to date for scoping reviews [43]. APC will be the first reviewer and guarantor of the review, whilst NGT and PM will be the second and third reviewers and assist APC in conducting elements of the review.

A scoping review was determined to be the most appropriate to address the aim of this research because this methodology seeks to provide an overview of the volume and distribution of the evidence base as well as to highlight where more research is warranted. Scoping reviews aim to synthesise evidence and assess the scope of literature on a topic, as well as to help evaluate whether a systematic review of the literature would be useful [42].

## 2.1. Identifying the review questions

We developed two key research questions to address our research aims:

1. When evidence-based practice is implemented for conservation and environmental management, what are the a) human and b) environmental outcomes?

2. Among these documents reporting these outcomes, what are the costs and ROI or VOI of evidence-based practice?

We use a definition of EBP based on Sackett et al. [3] and the Sicily statement on evidence-based practice [4]: *EBP involves making decisions and carrying out actions based on the best available, current, valid, and relevant evidence. Decisions and actions are also shaped by the subjects to which they apply, informed by the tacit and explicit knowledge of those implementing them, and within the context of available resources.*

Our research question formulation was guided by item 4 in the PRISMA-ScR scoping review extension checklist (see S3 Appendix) [42]. Through the PCC framework, we identified the main elements (Population or participants/Concept/Context), which helped to conceptualize the review questions and the study objectives. The JBI recommends the use of the PCC [44] framework to identify the main concepts for the primary review questions and to inform the search strategy. S1 Appendix includes the PCC framework along with the identified keywords for developing search terms.

## 2.2. Identifying potentially relevant documents

We will conduct systematic searches in Web of Science [Core Collection], CAB Abstracts, Scopus, GreenFILE, Engineering Village (Inspec and Compendex), ProQuest Dissertations & Theses, and EBSCOhost Business Source Ultimate to gain broad coverage across fields of conservation and environmental management and to avoid missing documents and thus an unrepresentative sample of the literature [45]. We will also search the general web-based search engine Google Scholar and use the R package gsscraper to bulk download the first 400 relevant references found because previous work has shown that this search engine can be useful for finding both academic and grey literature [46]. Furthermore, we will also query the Google search engine and other specific grey literature repositories including British Ecological Society's Applied Ecology Resources and Overton, downloading the first 400 results from each as well. We will add further search engines and repositories for accessing grey literature during this review and document any searches accordingly. For example, EThOS (a British Library database for UK theses and dissertations) was unavailable at the time of writing due to a cyber-attack.

Table 1 provides examples of preliminary database searches conducted including details of the date ranges searched and the search fields used with search terms shown in S2 Appendix. The Web of Science Core Collection accessed via Imperial College London's institutional access includes the following editions: Science Citation Index Expanded

**Table 1. Preliminary database search results and parameters for each bibliographic platform or database. Search terms are reported in S2 Appendix. We will update searches after publication of this protocol – date range searched is currently: 1st Jan 1992 to 15th Jan 2025.**

| Bibliographic platform or database | Search details | Results |
|---|---|---|
| Scopus | Field: Title, keyword and abstract (TITLE-ABS-KEY) | 13665 |
| Web of Science Core Collection | Field: Topic (TS) | 7806 |
| CAB Abstracts | Field: Topic (TS) | 6380 |
| Engineering Village | Field: subject, title and abstract (KY) | 5184 |
| GreenFILE | Field: All text (TX) | 1282 |
| ProQuest Dissertations & Theses Global | Field: Summary | 731 |
| EBSCOhost Business Source Ultimate | Field: Abstract | 587 |

(SCI-EXPANDED)--1970-present, Science Citation Index Expanded (SCI-EXPANDED)--1970-present, Social Sciences Citation Index (SSCI)--1970-present, Social Sciences Citation Index (SSCI)--1970-present, Arts & Humanities Citation Index (AHCI)--1975-present, Arts & Humanities Citation Index (AHCI)--1975-present, Conference Proceedings Citation Index – Science (CPCI-S)--1990-present, Conference Proceedings Citation Index – Science (CPCI-S)--1990-present, Conference Proceedings Citation Index – Social Science & Humanities (CPCI-SSH)--1990-present, Conference Proceedings Citation Index – Social Science & Humanities (CPCI-SSH)--1990-present, Emerging Sources Citation Index (ESCI)--2015-present, Emerging Sources Citation Index (ESCI)--2015-present.

We will search for documents from 1992 to 2025 without any geographical restriction and document the exact search dates in the final review. We chose 1992 as the earliest publication year because this is when the terms evidence-based medicine (EBM) and evidence-based practice (EBP) were first introduced in the medical literature, widely regarded as the first discipline to promote EBP [1].

We will use the R package litsearchr [47] to identify potentially useful additional keywords based on those found in the documents during the preliminary literature search – the first reviewer will manually review suggested keywords and add those considered to be useful to the final search. The search string currently includes keywords related to: evidence-based practice, evaluations and tests, impacts or effects, ROI or VOI, and conservation and environmental management (S1 and S2 Appendices). We use truncation ('*'), wildcards (e.g., '*' and '?'), and Boolean operators (AND/ OR), as appropriate, to form the specific search terms for each bibliographic platform and database's rules and configurations (S2 Appendix). Librarians at Imperial College London were consulted for the development of the search strings for all databases.

### 2.3. Abstract screening and stopping criteria

Following the search, we will collate all identified records from the different databases into Mendeley citation manager and remove duplicates using the R package revtools [48]. We will upload the references to ASReview, an open-source tool that uses AI-assisted 'researcher-in-the-loop' screening procedures. We intend to use an augmented approach to using ASReview, which will allow us to efficiently speed up the screening process using AI whilst also being as transparent and accurate as possible. This augmented approach will help to identify potentially relevant documents that a human expert alone might exclude erroneously and avoid some of the current limitations of ASReview. This will also encompass the core steps in the SYstematic review Methodology Blending Active Learning and Snowballing (SYMBALS) [49], which we apply to both the abstract screening (active learning to limit the impact of false positives on efficiency) and full-text screening (snowballing to limit the impact of false negatives on comprehensiveness) stages of this review. The combination of applying the SYMBALS methodology to the ASReview tool has also been shown to outperform other methods such as FAST[2] both in terms of efficiency and recall [49]. We will also download a data file from the ASReview tool that documents all decisions made during the screening process to ensure full transparency and reproducibility [50]. We will use ASReview LAB v2.0a8 [51], or a more recent version if one becomes available – this will be documented in the final review [51].

The basic process of ASReview that we intend to augment is an active learning pipeline with five main elements: 1. Feature Extractor algorithm (embedding model), 2. Classification algorithm, 3. Balancing strategy, 4. Query Strategy, and 5. Stopping Rule. This active learning approach uses continuous ranking of references derived from assessments of abstracts by the author as being potentially relevant or irrelevant, following the eligibility criteria. ASReview is considered accurate and efficient in other reviews that have tested this tool's performance [50,52–58], however, we intend to augment each of these five elements to improve the rigour and comprehensiveness of our machine learning-assisted review. This is possible thanks to open source nature of ASReview and its built-in functionality to enable custom extensions.

First, instead of the ASReview's basic TF-IDF feature extractor, we will implement the ModernBERT Embed model, which is an embedding model from the highly effective family of BERT (Bidirectional Encoder Representations from Transformers) models. This model can process sequences up to 8,000 characters, sufficient for abstracts of varying lengths,

whilst ASReview's other feature extractor choice (S-BERT All-mpnet-base-v2 embedding model) can only handle 384 characters and might be insufficient for some longer abstracts. The ModernBERT Embed model, like S-BERT models, is specialised in evaluating sentences, allowing us to map abstracts to 768 different semantic dimensions and use these to find similar papers to the relevant papers we identify.

Second, we will use the ModernBERT Embed model in combination with a logistic regression classifier (instead of the ASReview default of Naïve Bayes), which has been shown to perform best in simulation documents in terms of recall and efficiency with BERT embedding models [55,59] – a logistic regression classifier performs particularly well for test datasets with a low proportion of relevant documents [60].

Third, we will also use dynamic resampling, a data balancing strategy [60], to decrease the imbalance in the data used to train the classifier, as we expect to find a very high ratio of irrelevant documents to relevant ones (typically 97:3 for medical and environmental topics [61,62] – i.e., relatively few positive data points). ASReview requires only at least one relevant and irrelevant article to be added as its 'prior knowledge' initially, but starting with an initially small number of 'seed documents' could lead to severe bias if such documents are selected in a biased way, even when using data balancing strategies – i.e., a high quality, representative labelled dataset is vital for machine learning tools to perform effectively [50]. Our initial searches using our preliminary search terms found three potential seed documents that were relevant (Table 2). Therefore, to tackle the data imbalance problem further and provide a larger, high-quality labelled dataset, we will also generate an additional 22 synthetic relevant documents (25 relevant seed documents in total) using a state-of-the-art Large Language Model such as Claude Sonnet 4.0 (depending on the best available model at the time of conducting our review, which will be documented in the final review). Similarly, we will randomly select 75 irrelevant seed documents from our initial searches (to achieve a 3:1 ratio for 100 seed documents in total for the training dataset). The first and second reviewers will check the relevant and irrelevant labels for the seed documents as an additional quality filter. This will involve checking the accuracy of the labels and that there is a wide representation of different topics, locations, experimental designs (i.e., good representativeness of examples), including clear positive and negative examples, as well as more marginal examples (i.e., potential 'edge cases' [50]). Any disagreements between reviewers in the seed documents selected will be discussed and corrected if necessary – either by regenerating extra synthetic documents or resampling the initial searches.

Fourth, we will use the default query strategy in ASReview, certainty-based sampling whereby the next article presented to be screened is the one with the next highest relevance score (i.e., moving down a ranked list in order of decreasing potential relevance [53]). Certainty-based sampling has been shown to perform well with weighted data balancing strategies [66]. Furthermore, our augmented balancing strategy (through generating synthetic seed documents), and quality filtering of our training set for our classifier discussed previously, will also mitigate potential bias from the selection of initial seed documents that can affect certainty-based sampling.

**Table 2. Preliminary seed or benchmark documents labelled as relevant.**

| References | Retrieved using preliminary search strings |
|---|---|
| McConnachie MM, Cowling RM. On the accuracy of conservation managers' beliefs and if they learn from evidence-based knowledge: A preliminary investigation. J Environ Manage. 2013;128: 7–14. https://doi.org/10.1016/J.JENVMAN.2013.04.021 [63] | Yes: Web of Science, Scopus, GreenFILE, Engineering Village. |
| Walsh JC, Dicks L V., Sutherland WJ. The effect of scientific evidence on conservation practitioners' management decisions. Conservation Biology. 2015;29: 88–98. https://doi.org/10.1111/COBI.12370 [64] | Yes: Web of Science, Scopus. |
| Santangeli A, Sutherland WJ. The Financial Return from Measuring Impact. Conserv Lett. 2017;10: 354–360. https://doi.org/10.1111/CONL.12284 [65] | Yes: Web of Science, Scopus, GreenFILE. |

Fifth, to do any machine learning-assisted review reliably, it is vital to define a stopping rule [67]. However, defining appropriate stopping rules is still a pervasive problem in systematic review methods and for which there is typically a general lack of useful guidance [67]. For example, it is common for arbitrary, inconsistent stopping rules to be used by review papers [67,68], such as once $n$ irrelevant papers screened in a row, or to use methodologies that appear to lack statistical justifications or proper empirical evaluations (e.g., SAFE procedure [69]). Instead, we will use a statistically justified method [70] based on point processes (statistical models to represent the occurrence of random events) whereby rate functions model the occurrence of relevant documents in the ranking and compares four candidates (hyperbolic, power law, exponential, and AP-Prior distribution). The required information is the target recall ($\ell = 95\%$ - as suggested by [71]), confidence levels that the recall has been achieved prior to stopping ($p = 95\%$), the total number of documents in the ranking (n), as well as the initial sample size ($\alpha = 5\%$ of n) and the sample increment size ($\beta = 2.5\%$ of n) that control the number of documents examined between each application of the model to check whether to stop. Each time the model runs (if there is sufficient data to fit the rate function), it will output the estimated rank at which that the target recall has been met with the desired level of confidence.

Once the first reviewer has screened enough documents to check the stopping rule, a second and third reviewer will check a random subset of screened documents to calculate inter-rater reliability compared using Cohen's Kappa [72]. The sample size of this subset will be determined using a power analysis with the R package kappaSize [73] (to test kappa0 = 0.4 vs. kappa1 = 0.6, with alpha = 0.05, prevalence of relevant studies = 0.05 and power = 0.80; approximately 500 documents). If Kappa values of at least 0.6 are found between the three reviewers (generally considered as high agreement), we will proceed to full-text screening and assume that the machine learning classifier was trained reliably by the first reviewer. If Kappa is less than 0.6, we will discuss any disagreements, document them, and correct them if necessary in ASReview. We will then rerun the model to estimate the rank at which we can reliably stop – if necessary, screening will continue until the stopping rule is met again. We will keep a log of excluded documents stating the reason for exclusion and report this using the PRISMA-ScR diagram for all stages of the review using the decision tree presented in Fig 1 and associated eligibility criteria.

## 2.4. Full-text screening and snowballing

We will access the full-texts of documents that meet the eligibility criteria from the abstract screening stage and upload them to SysRev – an online tool that allows for screening and data extraction by review teams [75]. Using SysRev also enables the full-text screening process, as for abstract screening in ASReview, to be as transparent as possible because those interested in the review can fully examine decisions about the inclusion/exclusion of documents [75]. The first reviewer will screen the full-text documents against the eligibility criteria and a random subset will also be screened by the second and third reviewers (using kappaSize [73] as per the abstract screening but with the prevalence of relevant articles = 0.5; approximately 90 documents). If Kappa is over 0.6, the relevant full-text documents will be carried forward for data extraction and summarisation in the scoping review. If Kappa is less than 0.6, we will discuss any disagreements, document them, and correct them if necessary, and repeat the process by adding a further 10 documents at a time until Kappa reaches at least 0.6. If we find any documents that should be excluded according to the eligibility criteria, we will remove them and report this in the PRISMA-ScR diagram.

Finally, we will conduct backward snowballing following [49] (considering reference lists of relevant documents), and additional forward snowballing (looking at documents citing relevant documents) for added comprehensiveness. This goes beyond the SYMBALS methodology requirements, which suggests just backward snowballing is typically sufficient [49]. We will use the R package citationchaser [76] to conduct snowballing and repeat the abstract screening stage described previously, using the same point process model stopping rule to decide when to end screening of the snowballed documents. Any relevant documents included from snowballing will be documented accordingly.

## 2.5. Eligibility criteria

We will include both primary and secondary published and grey literature. Fig 1 details the decision tree we will use to screen the documents.

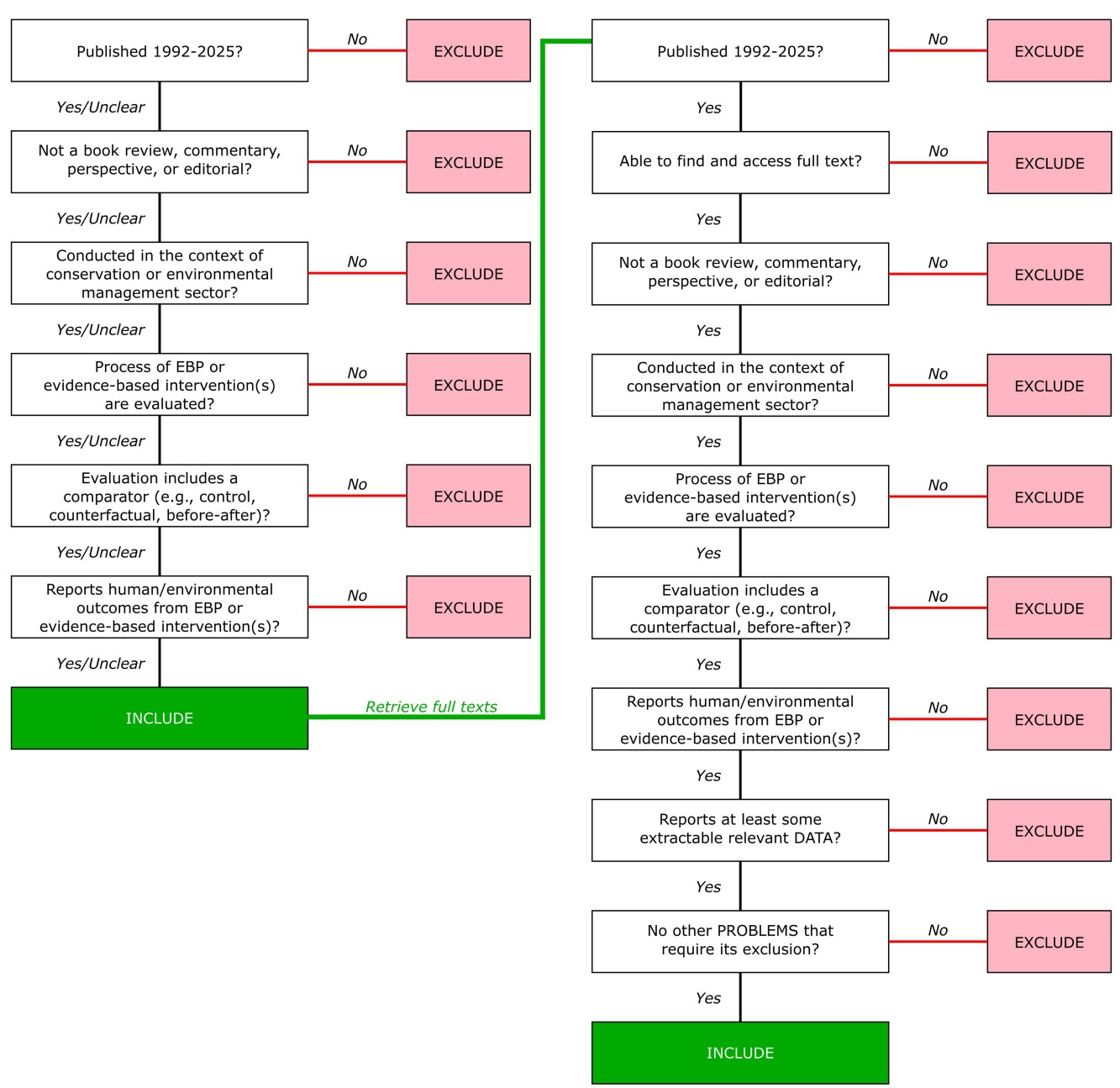

**Fig 1. Decision tree aid for screening abstracts and full texts – adapted from [ 74].**

Our specific inclusion criteria are that the article describes implementing an EBP change (e.g., starting or stopping EBP) in conservation or environmental management and measures the resulting outcomes against a comparator. The outcomes could be human (e.g., measuring decision-makers' behaviour, beliefs, or attitudes) or environmental (e.g., measuring the impact an EBP change had on the conservation of a species, erosion of a coastline, or on water quality). This includes, for example, the implementation of an explicitly evidence-based or evidence-informed intervention and comparing the outcomes to an appropriate comparator. It also includes providing evidence and measuring how that affects decision-making compared to a comparator without evidence. We will only be able to include documents that explicitly state that evidence has been used (using our definition of evidence in the introduction from [27]) to inform practice or explicitly states that an evidence-based/informed intervention has been evaluated – the type(s) of evidence used will be documented in the data extraction stage too. Reporting of EBP costs, ROI or VOI is not part of the eligibility criteria; these data will be extracted from all eligible documents that report them.

We will include documents written in non-English languages – those in non-English languages will be translated using DeepL, an accurate AI-driven translation tool [77]. However, we will not search in non-English languages (see Section 3 Limitations). Grey literature available in the public domain from websites (in the form of reports etc.) will be considered for the review.

Accordingly, our exclusion criteria are as follows: documents published prior to 1992; book reviews, commentaries, perspectives, or editorials; not on conservation or environmental management; not evaluating process of EBP or an evidence-based intervention(s); no comparator; no human or environmental outcomes reported. At the title/abstract screening stage, we will retain any documents that meet all eligibility criteria or it is unclear whether they do or not for full-text screening (Fig 1).

All eligibility criteria will be discussed between the reviewers prior to and during screening checks and any changes justified and documented in the final review.

### 2.6. Extracting and summarising data

The first reviewer will perform data extraction in SysRev to extract the following data: outcomes of study (human or environmental, specific metrics used, and summary of findings), country of study, study system or habitat (if appropriate, using IUCN classification), type of publication (e.g., primary or secondary, published or grey literature), year the study was published, EBPs implemented (e.g., literature search, assessment of evidence), evidence types used (e.g., published literature, grey literature, local, practitioner, or Indigenous knowledge), any costs of EBPs numerical or qualitative (e.g., staff time, training costs, if stated), return or value on investment and its method of calculation (ROI or VOI and the type of economic assessment used, if measured) of EBPs, methodological approach (specific experimental or observational study design – e.g., Before-After, Randomised Experiment), type of organisation involved in EBP (e.g., practitioner, policymaker, academic). The categories of qualitative data collected will be refined iteratively depending on the contents of the eligible documents reviewed. We will present all results in table or graphical format and summarise them in text format. Finally, a narrative summary of the review findings will be provided and organized into thematic categories relating to types of EBP changes, their outcomes (human vs environmental), key findings and research gaps, using frequencies and qualitative thematic analysis [78].

### 2.7. Institutional review board statement

Not Applicable. This study will review already published and publicly available data retrieved from the databases mentioned previously.

### 3. Limitations

Although our scoping review's methods are designed to be as comprehensive as possible, there is the potential that we will miss a small percentage of relevant documents (5% [49,71,79]) that are not accessible, that use terminology not

captured in our search strings, and from stopping the search at a certain point. Another limitation will be that we will be mostly searching English language literature given that we will not explicitly search using non-English language terms or in non-English language journals (although we will not exclude any non-English language documents found through the bibliographic platforms, databases, and search engines stated earlier). Furthermore, methodology quality appraisals and assessments of the documents and evidence included in the scoping review will not be undertaken as in a traditional systematic review, which is a potential limitation – although Arksey and O'Malley [80] and Pollock et al. [81] state that quality appraisals of evidence is not the focus of scoping reviews as this is usually completed in a systematic review that might seek to build on this scoping review if sufficient literature is found. Our review will also only consider the published and grey literature and will not incorporate other forms of evidence, such as practitioner or local knowledge, that might provide information on the impact and ROI of EBP. Nevertheless, a core aim of this review is to show whether there is a clear research gap in the published and grey literature that needs to be filled by primary research, such as conducting high quality impact evaluations and social science research to collect and share insights on the impact of using evidence on conservation outcomes to underpin key assumptions of EBP.

## 4. Conclusions

This review is the first to our knowledge to comprehensively report on the literature testing whether evidence use and EBP leads to better outcomes in conservation and environmental management. Its main aim is to understand to what extent the impact and ROI of EBP is studied in the context of conservation and environmental management. The findings from our scoping review will be useful to those implementing evidence-based conservation, and EBP more generally, to understand where there may be gaps in our understanding of how evidence use may improve decision-making and ultimately practice and policy. This is highly important for reliably informing practitioners and decision-makers, both those resistant or receptive to adopting EBP, as to its benefits and costs and its potential to improve practice and policy.

## Supporting information

**S1 Appendix. PCC Framework used to construct search strings**
(DOCX)

**S2 Appendix. Preliminary search terms**
(DOCX)

**S3 Appendix. PRISMA-ScR scoping review extension checklist**
(DOCX)

**S1 Checklist. PRISMA-P SystRev checklist.**
(DOCX)

## Author contributions

**Conceptualization:** Alec P. Christie, Philip A. Martin, Nigel G. Taylor.

**Data curation:** Alec P. Christie.

**Formal analysis:** Alec P. Christie.

**Funding acquisition:** Alec P. Christie.

**Investigation:** Alec P. Christie, Philip A. Martin, Nigel G. Taylor.

**Methodology:** Alec P. Christie, Philip A. Martin, Nigel G. Taylor.

**Project administration:** Alec P. Christie.

**Validation:** Alec P. Christie, Philip A. Martin, Nigel G. Taylor.

**Visualization:** Alec P. Christie.

**Writing – original draft:** Alec P. Christie, Philip A. Martin, Nigel G. Taylor.

**Writing – review & editing:** Alec P. Christie, Philip A. Martin, Nigel G. Taylor.

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
