## [Decision Letter · Decision Letter 0]

PONE-D-25-03610The impact and return-on-investment of evidence-based practice in conservation and environmental management: a machine learning-assisted scoping review protocolPLOS ONE

Dear Dr. Christie,

Thank you for submitting your manuscript to PLOS ONE. After careful consideration, we feel that it has merit but does not fully meet PLOS ONE’s publication criteria as it currently stands. Therefore, we invite you to submit a revised version of the manuscript that addresses the points raised during the review process. The proposed scoping review will provide important insight into the value and impacts of evidence-based practice in environmental management and conservation. Although the proposed protocol appears sound, both reviewers made excellent suggestions that would strengthen the rationale and methodology.

Please address each of their specific questions and comments, ensuring that your revisions include:

A comparison EBP in medicine and conservation/environmental management, highlighting challenges/aspects unique to the latter context.Elaboration on what constitutes evidence in a conservation context and incorporate into your methodology for screening/study selection.Text related to the potential limitations of relying on literature to assess the extent to which evaluations of impacts and ROI of EBP in environmental/conservation management are conducted for human and environmental outcomes.

We look forward to receiving your revised manuscript.

Kind regards,

Jenilee Gobin

Academic Editor

PLOS ONE

Journal Requirements:

2. In your cover letter, please confirm that the research you have described in your manuscript, including participant recruitment, data collection, modification, or processing, has not started and will not start until after your paper has been accepted to the journal (assuming data need to be collected or participants recruited specifically for your study). In order to proceed with your submission, you must provide confirmation.

4. Please remove all personal information, ensure that the data shared are in accordance with participant consent, and re-upload a fully anonymized data set.

Reviewers' comments:

Reviewer's Responses to Questions

**Comments to the Author**

1. Does the manuscript provide a valid rationale for the proposed study, with clearly identified and justified research questions?

Reviewer #1: Yes

Reviewer #2: Yes

2. Is the protocol technically sound and planned in a manner that will lead to a meaningful outcome and allow testing the stated hypotheses?

Reviewer #1: Yes

Reviewer #2: Yes

3. Is the methodology feasible and described in sufficient detail to allow the work to be replicable?

Reviewer #1: Yes

Reviewer #2: Yes

4. Have the authors described where all data underlying the findings will be made available when the study is complete?

Reviewer #1: Yes

Reviewer #2: Yes

5. Is the manuscript presented in an intelligible fashion and written in standard English?

Reviewer #1: Yes

Reviewer #2: Yes

6. Review Comments to the Author

You may also provide optional suggestions and comments to authors that they might find helpful in planning their study.

Reviewer #1: General comments

The overall aim of this study is to assess the impact and return on investment (ROI) of evidence-based practice (EBP) in conservation and environmental management through a scoping review. This is an important and timely study, as it seeks to fill a knowledge gap in evaluating the outcomes of EBP beyond healthcare, particularly in the environmental sciences. The concepts are well introduced in the Introduction section, with a clear rationale provided for the study. The Material and Methods section is detailed and adheres to established scoping review methodologies, including the use of PRISMA-ScR guidelines and AI-assisted screening tools (ASReview, FAST2). The Results are not yet available, as this is a protocol paper, but the proposed approach appears comprehensive. The discussion effectively highlights the potential significance of the study and its contributions to conservation decision-making.

However, there are a few areas that require further clarification and refinement. I, therefore, advise for a minor revision of the manuscript, including improvements in methodological transparency and addressing potential limitations. I provide specific comments below that I hope will be useful in improving the manuscript.

Specific comments

The introduction clearly establishes the importance of EBP; however, it would benefit from a more structured comparison between the conservation and healthcare sectors in terms of evidence-based decision-making, as well as a more specific definition of EBP in the context of conservation. It is also related to the inclusion/exclusion criteria of your scoping review: How will you define that a study explicitly used some form of evidence? Won’t you miss concrete applications of conservation science evidence in policy and everyday management actions by using literature only? Don’t we also need interviews with conservation practitioners to have a sense of the use of EBP in conservation? Perhaps consider mentioning this as a perspective in your review?

The use of machine learning tools is innovative, but further details on their added value in systematic reviews would strengthen the methodological justification. What will they bring concretely, beyond saving some of your time?

The right panel of Figure 1 is confusing due to the use of general terms such as ‘LANGUAGE’ and ‘TYPE’, which are not specified.

The authors acknowledge that scoping reviews do not assess the quality of included studies. However, given the study’s emphasis on ROI, will there be any considerations regarding the methodological rigor of economic assessments in conservation?

I thank the authors and editor for this interesting reading. As always, I sign this review and welcome the authors to engage with me on this topic. I’d be interested in learning more about your project and its future results.

Best regards,

Simon Lhoest

Reviewer #2: General Comments:

This protocol aims to understand the state of the evidence regarding impact and ROI of evidence-based practice in conservation and environmental management. The scoping review approach it describes is both clear and thorough. I think this work will make an important contribution to the field, which needs to better understand the evidence for, or lack thereof, impacts from evidence-based practice.

One other point I think that is worth mentioning in this protocol is how EBP is different in conservation vs. medicine. I think it should be pointed out that the nature of conservation interventions often do not lend well to randomized control trials. This does not mean we should give up or not try, but it does raise the issue of other approaches to evaluation to build the evidence base, such as counterfactuals; I think this notion should be mentioned somewhere in this paper and also likely should be a keyword for searches.

Specific Comments:

Abstract: Line 17: There is an extra “appears to” here before “be”

Lines 47-48: Consider providing references for the other “evidence-based” terms you list

Lines 51-52: What about the goal of more effective outcomes? This is perhaps the most important reason for EBP

Lines 71-72: In regards to not including “Indigenous knowledge in conservation and environmental management [24–26]” - this is an important shortcoming of some of the key systematic review guidance and I am pleased to see it highlighted here. The main stream guidance is typically lacking in providing guidance for including these other types of evidence. Some have pushed for a different terminology as a way to call attention to the need to include these other forms of evidence; could be worth pointing this out (e.g., Sterling et al. 2017 discusses using the broader more inclusive “evidence-informed” terminology instead of the more narrow “evidence-based” conclusion framing)

Lines 81-82: Seems here could point out how conservation interventions often vary in contexts, making it difficult to do randomized trials, etc. especially in comparison to say medicine

Lines 278-281: Should this also list change in cost as a human impact?

7. PLOS authors have the option to publish the peer review history of their article (what does this mean? ). If published, this will include your full peer review and any attached files.

**Do you want your identity to be public for this peer review?** For information about this choice, including consent withdrawal, please see our Privacy Policy .

Reviewer #1: **Yes:** Simon Lhoest

Reviewer #2: No

---

## [Author Response · Author response to Decision Letter 1]

20 May 2025

Please see attached response to reviewers document.

---

## [Decision Letter · Decision Letter 1]

The impact and return-on-investment of evidence-based practice in conservation and environmental management: a machine learning-assisted scoping review protocol

PONE-D-25-03610R1

Dear Dr. Christie,

We’re pleased to inform you that your manuscript has been judged scientifically suitable for publication and will be formally accepted for publication once it meets all outstanding technical requirements.

Kind regards,

Jenilee Gobin

Academic Editor

PLOS ONE

Additional Editor Comments (optional):

Reviewers' comments:

Reviewer's Responses to Questions

**Comments to the Author**

1. Does the manuscript provide a valid rationale for the proposed study, with clearly identified and justified research questions?

Reviewer #1: Yes

2. Is the protocol technically sound and planned in a manner that will lead to a meaningful outcome and allow testing the stated hypotheses?

Reviewer #1: Yes

3. Is the methodology feasible and described in sufficient detail to allow the work to be replicable?

Reviewer #1: Yes

4. Have the authors described where all data underlying the findings will be made available when the study is complete?

Reviewer #1: Yes

5. Is the manuscript presented in an intelligible fashion and written in standard English?

Reviewer #1: Yes

6. Review Comments to the Author

You may also provide optional suggestions and comments to authors that they might find helpful in planning their study.

Reviewer #1: Thanks for this updated manuscript in which all my comments have been appropriately addressed.

7. PLOS authors have the option to publish the peer review history of their article (what does this mean? ). If published, this will include your full peer review and any attached files.

**Do you want your identity to be public for this peer review?** For information about this choice, including consent withdrawal, please see our Privacy Policy .

Reviewer #1: **Yes:** Simon Lhoest

---

## [Editor Report · Acceptance letter]

PONE-D-25-03610R1

PLOS ONE

Dear Dr. Christie,

I'm pleased to inform you that your manuscript has been deemed suitable for publication in PLOS ONE. Congratulations! Your manuscript is now being handed over to our production team.

Kind regards,

on behalf of

Dr. Jenilee Gobin

Academic Editor

PLOS ONE